# The Impact of Accumulating Herbage Masses in Autumn on Perennial Ryegrass Sward Characteristics

Caitlin Looney [1,2], Astrid Wingler [2] , Daniel Donaghy [3] and Michael Egan [2,*]

1   Teagasc, Animal & Grassland Research and Innovation Centre, Moorepark, Fermoy, P61 C996 Cork, Ireland; caitlin.looney@teagasc.ie

2   School of Biological, Earth and Environmental Sciences and Environmental Research Institute, University College Cork, T23 TK30 Cork, Ireland; astrid.wingler@ucc.ie

3   School of Agriculture and Environment, Massey University, Private Bag 11 222, Palmerston North 4442, New Zealand; d.j.donaghy@massey.ac.nz

*   Correspondence: michael.egan@teagasc.ie

**Abstract:** Autumn grazing management aims to accumulate herbage for defoliation prior to a decrease in growth rates for the extension of the grazing season. The current study investigated the impact of building different target herbage masses (THMs) in autumn and imposing one of three different defoliation dates (DDs) between mid-October and late November on light transmitted to the base of the sward, free leaf lamina (FLL), leaf stage and internode elongation. Four THMs (low, medium, high and very high) and three DDs (DD1—15 October, DD2—7 November and DD3—21 November) were assigned to a 4 × 3 split plot design over two years. Light transmitted to the base of the sward was greatest in the low THM and decreased in all other THMs. Internode elongation increased in tillers in the medium to the very high THMs. Defoliation of the medium, high and, in particular, the very high THMs earlier in autumn reduced the effect of decreased light transmission on internode elongation. This study highlights that, as light transmitted to the base of the sward decreases, internode elongation increases, and this could negatively impact sward structure.

**Keywords:** autumn; defoliation date; target herbage mass; light transmission; free leaf lamina; internode elongation

## 1. Introduction

Perennial ryegrass (*Lolium perenne* L.; PRG) is the preferred species in temperate pasture-based systems [1] as it has the ability to produce large quantities of high-quality herbage [2], as well as being able to withstand intensive grazing [3]. Previous research [4] identified increasing the proportion of grazed grass in the diet of animals as an important objective of pasture-based systems, in addition to improving the environmental and economic sustainability within the farm system [5–7]. One way to increase the proportion of grazed grass in the diet is the extension of the grazing season, which has been reported to increase overall farm profitability by reducing feed and labour costs [6] and has been shown to be correlated with the reduction in the carbon footprint of milk [8]. Currently, the average length of the grazing season in Ireland is 223 days [9], with the aim to increase this to 300 days [10].

Grassland productivity is limited by low temperatures and seasonality [11] with lower amounts of herbage accumulating from autumn and over winter due to restrictive environmental conditions [12,13] compared to the summer period. To facilitate the extension of the grazing season, specific grazing management strategies are used; herbage masses are 'built and carried' [13] from August to the target grazing dates for the final grazing rotation (from approximately 15 October to 21 November) [14]. Herbage mass must be accumulated in autumn when growth rates allow for an extended grazing period [14,15]. Autumn grazing management studies to date have focused on animal productivity, including an increase in

milk protein concentration [15,16], a reduction in supplementation requirement [9,15] and the impact of autumn defoliation [17]. However, few studies have investigated the impact of autumn grazing management practices on sward morphology.

Previous studies have demonstrated that grazing PRG swards at the 'three-leaf stage' [18] and approximately 1500 kg DM ha$^{-1}$ [19] of herbage pre grazing is optimal for the maximum utilisation of high-quality herbage and allows sufficient build-up of water-soluble carbohydrates for PRG swards. The PRG plant is in the vegetative stage in autumn and has one actively growing leaf at a time, with three green leaves in total [20]. Once the plant moves above the three-leaf stage, the oldest leaf (first leaf to appear after defoliation) begins to senesce, which results in an increase in the proportion of senescent material in the sward [20,21]. To accumulate herbage masses, the defoliation interval is extended, which can result in more than three leaves being produced prior to defoliation, and this increases the number of leaves senesced [20]. Ultimately, leaves are continuously being produced and senescing, even after the ceiling yield is achieved [21]. This can result in no further accumulation of herbage, just an increase in senescent material at the base of the sward. Increased herbage and senescent material can reduce light available to the base of the sward [22]. Increasing the number of leaves past the three-leaf stage, as well as shading, can reduce the photosynthetic activity of the PRG leaves [23]. Previous studies on carrying high herbage masses over winter have found that the level of senescent material in the sward increases, and have concluded that swards should be grazed at a three-leaf stage [18]. However, limited research has been carried out on the implications of herbage accumulation in autumn on sward characteristics. As discussed above, the longer a sward remains undefoliated, the greater the chance of increased shading at the base of the sward. Studies have found shading can reduce tiller production and hinder tiller establishment and survival [24–26]. The resulting reduction in tiller production can reduce herbage production through decreasing the number of growing points in the sward [27]. An increase in shading at the base of the sward can also lead to the internode elongation of normally un-elongated internodes, where the distance between nodes increases [28], and the growing point is now no longer at ground level, which may have negative impacts on sward morphology such as increased tiller mortality.

The objective of this study was to investigate the effect of accumulating varying herbage masses in autumn and imposing different defoliation dates in the final grazing rotation on sward morphology, particularly leaf stage, lamina length and the presence of internode elongation.

## 2. Materials and Methods

### 2.1. Site

A plot trial was conducted at the Teagasc, Animal and Grassland Research and Innovation Centre, Moorepark, Fermoy, Ireland (Latitude 50°07′ N, Longitude 08°16′ W) from 1 August 2018 to 1 April 2019 (year 1) and 1 August 2019 to 1 April 2020 (year 2) [27]. The soil type was a free-draining acid brown earth of sandy to loam texture. Soils had a pH of 6.4, a phosphorus (P) index of 4 and a potassium (K) index of 3 (±0.8; scale of 1 to 4; 1 = deficient, 4 = no response to application of nutrient) [29]. The experimental site was of south facing aspect and located approximately 40 m above sea level. Swards were made up of PRG (*Lolium perenne* L.; >85%; seed mixture—3-way mix of cv. Astonenergy, Oakpark and Meiduno) predominantly, with the remaining 15% made of mainly annual weed grass (*Poa annua* L.) and broadleaf plants. Meteorological data were recorded at the experimental site over the experimental period (October to February). Average daily air temperature (°C), soil temp to a depth of 100 mm (°C), total daily rainfall (mm) and average solar radiation are shown for the measurement periods in each experimental year [30].

### 2.2. Experimental Design

The experimental design was a 4 × 3 split plot design with four replicates, resulting in 48 plots (1.5 m × 6 m). Four target herbage masses (THMs; low—500 kg DM ha$^{-1}$,

medium—1500 kg DM ha$^{-1}$, high—2000 kg DM ha$^{-1}$ and very high—3000 kg DM ha$^{-1}$; see Table 1 for achieved herbage masses) were targeted for accumulation on 15 October. The THMs were assigned to the whole plots, and then, 3 different autumn DDs were assigned to the 3 split plots within each whole plot. The growth rates achieved were higher (Table 1), particularly in the low THM, due to the compensatory growth experienced post drought in 2018. Three defoliation dates (DDs; DD1—15 October, DD2—7 November and DD3—21 November), chosen to reflect Irish autumn grazing management practices [14], were examined within the four THMs. Measurements of sward characteristics (see below) were carried out on each autumn DD (DD1, DD2 and DD3), in addition to the measurement of herbage mass and light at the base of the sward for the 1st defoliation in spring (21 February). Outside of the experimental period, plots were defoliated, using an Etesia mower (Etesia UK Ltd., Warwick, UK), to a residual of 4 cm when they reached a pre-grazing herbage mass of 1500 kg DM ha$^{-1}$ between February and July. In August, plot defoliation was tailored by adjusting the defoliation date and performing back-calculation from the previous daily growth rates from that time of year at the experimental site, to accumulate each THM for 15 October; the previous defoliated dates for the low, medium, high and very high THMs were 9 October, 19 September, 9 September and 20 August in year 1, and 8 October, 16 September, 5 September and 13 August in year 2, respectively. Over the experimental period, nitrogen was applied in early August at a rate of 35 kg N ha$^{-1}$ in the form of calcium ammonium nitrate (CAN; 27% N). Phosphorus and potassium were applied in January at 30 kg ha$^{-1}$ of 0-7-30. At the end of January, nitrogen was applied at a rate of 28 kg N ha$^{-1}$ in the form of urea (46% N). Outside of the experimental period, nitrogen was applied, post defoliation, at 20 kg N ha$^{-1}$ of CAN. In year 2, 30 kg ha$^{-1}$ of 0-7-30 was applied to all plots in August.

**Table 1.** Effect of target herbage mass (low—500 kg DM ha$^{-1}$, medium—1500 kg DM ha$^{-1}$, high—2000 kg DM ha$^{-1}$ and very high—3000 kg DM ha$^{-1}$) at defoliation on the herbage mass (kg DM ha$^{-1}$), light at the base of the sward (%), leaf stage (leaves), free leaf lamina (cm) and the total green leaf lamina (cm).

| | Target Herbage Mass [1] | | | | | |
|---|---|---|---|---|---|---|
| **Variable** | **Low** | **Medium** | **High** | **Very High** | **S.E.M** | **Significance** |
| Herbage mass (kg DM ha$^{-1}$) | 900 [a] | 1735 [b] | 1938 [b] | 2917 [c] | 153.6 | ***[2] |
| Light at the base of the sward (%) | 66 [a] | 57 [b] | 58 [b] | 55 [b] | 2.8 | *** |
| Leaf stage (leaves) | 2.2 [a] | 2.8 [b] | 3.3 [c] | 3.9 [d] | 0.06 | *** |
| Free leaf length (cm) | 18.6 [a] | 24.9 [b] | 27.4 [c] | 32.8 [d] | 0.41 | *** |
| Total green leaf lamina per plant (cm) | 36.3 [a] | 52.2 [b] | 58.7 [c] | 68.2 [d] | 1.51 | *** |

Different superscripts within rows denote significantly different means ($p < 0.05$). [1] mean data across the three defoliation dates. ***[2]—level of significant—*** = 0.001.

### 2.3. Herbage Mass

The entire plot was harvested using the Etesia mower (Etesia UK Ltd., Warwick, UK) to a post cutting height of >3.5 cm (targeted residual). The mown herbage from the plot was collected and weighed. A sample of approximately 300 g was collected from each cut strip. A subsample of 100 g was weighed and dried for 16 h at 60 °C to determine dry matter (DM) content.

### 2.4. Light at the Base of the Sward

The photosynthetically active radiation (PAR; 400–700 nm) was measured on all plots pre and post each defoliation using a line quantum sensor (1 m with 10 interval sensor points; LI-COR Inc. Lincoln, NE, USA) attached to a data logger. The above-canopy PAR was recorded for each plot first by placing the line quantum sensor at canopy level beside the plot, with nothing interfering with its light interception, and a ten-second average from across the ten points of the line quantum sensor was recorded. The below-canopy PAR was

taken, twice in each plot, by inserting the line quantum sensor at the base of the canopy, making sure not to disrupt the sward. For each plot, an average of both below-canopy PARs was used. The percentage of light transmitted to base of the sward was calculated using the equation below:

(Below-canopy measurement)/(above-canopy measurement) × 100 = % light at base of sward

### 2.5. Leaf Stage

The perennial ryegrass leaf stage was measured by taking ten vegetative PRG tillers randomly selected from each plot prior to defoliation at each DD in autumn and again in spring, as outlined by Dairy NZ [31]. On each tiller, the remnant leaf was identified (if more than one remnant leaf was present, only the uppermost remnant leaf was included). The length of this remnant leaf (both lamina and sheath) was compared with the leaf above it (first new leaf), and if it was less than half the size of the new leaf, it was not included in the measurement; if it was greater than half the size, it was counted as half, three quarters or a whole leaf in its own right. Each fully grown leaf, un-defoliated and with a pointed tip, was counted. If the uppermost leaf was not fully grown, it was compared in size to the leaf below it, and counted to the nearest ¼ of a leaf. The leaf stage of each of the individual tillers was recorded and averaged to give the leaf stage for each plot.

### 2.6. Free Leaf Lamina, Total Green Leaf Lamina Length and Internode Elongation

From each plot, prior to defoliation, 45 mature PRG tillers with no visible daughter tillers were manually harvested at the ground level using a scalpel blade at random areas within each plot. On each harvested tiller, the extended tiller height (ETH) was measured using a graduated ruler from ground level to the tip of the longest green leaf lamina. The extended sheath height (ESH) was measured from ground level to the collar of the newest sheath. The free leaf lamina (FLL) was then calculated by subtracting the ESH from the ETH [19]. The total green leaf lamina length was also recorded by adding up the length of all present (non-senescent) leaves (lamina only). After the ESH and ETH were recorded, the sheath was carefully removed and examined for internode elongation. If internode elongation was visible, the length of elongation was measured to the nearest mm from the location of the apical meristem.

### 2.7. Statistical Analysis

Statistical analyses were carried out using SAS 9.4 (SAS Institute Inc., Cary, NC, USA, 2002). A Tukey test was used to compare every mean with all other means and account for the scatter of all groups. For the % internode elongation means, a Tukey–Kramer test was run to account for unequal sample sizes. The effect of THM and DD on herbage mass, light at the base of the sward, leaf stage, free leaf lamina, total green leaf lamina length, internode elongation and the no. of nodes was determined using the PROC MIXED procedure in SAS, with THM, DD, year and plot being used in the model. All non-significant interactions were removed from the model. Plot was the experimental unit and year was the random factor. Data are presented as least square means ± standard error. Variables were analysed using the following model:

$$Y_{jkl} = \mu + \llbracket THM \rrbracket_j + \llbracket DD \rrbracket_k + \llbracket year \rrbracket_l + (\llbracket THM \rrbracket_j \times \llbracket DD \rrbracket_k) + (\llbracket year \rrbracket_l \times \llbracket THM \rrbracket_j) + (\llbracket year \rrbracket_l \times \llbracket DD \rrbracket_k) + (\llbracket DD \rrbracket_k \times \llbracket THM \rrbracket_j) \times e_{jkl}$$

where:

$\mu$ = mean value for the variable;

$e_{jkl}$ = residual error term;

$Y_{jkl}$ = target herbage mass (kg DM ha$^{-1}$), light at the base of the sward (%), leaf stage (leaves), free leaf lamina (cm), total green leaf lamina length (cm), percentage of tillers

with visible internode elongation (%), length of internode elongation (cm) and percentage of tillers with an apical meristem >4 cm.

## 3. Results

### *3.1. Herbage Mass*

The interaction between DD and year was significant ($p < 0.001$). In year 1, DD2 ($2330 \pm 108.1$ kg DM ha$^{-1}$) and DD3 ($2402 \pm 108.1$ kg DM ha$^{-1}$) had a greater herbage mass than DD2 ($1800 \pm 108.6$ kg DM ha$^{-1}$) and DD3 ($1547 \pm 108.6$ kg DM ha$^{-1}$) in year 2, respectively. There was a significant ($p < 0.001$) effect of THM on herbage mass; the very high THM had a greater herbage mass than all other THMs. The medium and high THMs were both greater than the low THM (Table 1). Defoliation date significantly ($p < 0.001$) affected herbage mass; the defoliation of swards in mid-October (DD1) resulted in a lower herbage mass compared to November (DD2 and DD3; Table 2). The total annual herbage mass was significantly ($p < 0.001$) greater in year 1 ($2172 \pm 62.7$ kg DM ha$^{-1}$) compared to year 2 ($1634 \pm 62.7$ kg DM ha$^{-1}$).

**Table 2.** The effect of defoliation date (DD; DD1—15 October, DD2—7 November and DD3—21 November) in autumn on the herbage mass (kg DM ha$^{-1}$), light at the base of the sward (%), leaf stage (leaves), free leaf lamina (cm) and the total green leaf lamina (cm) are presented.

| | Defoliation Date [1] | | | | |
|---|---|---|---|---|---|
| **Variable** | **DD1** | **DD2** | **DD3** | **S.E.M** | **Significance** |
| Herbage mass (kg DM ha$^{-1}$) | 1578 [a] | 2100 [b] | 1940 [b] | 153.6 | ***[2] |
| Light at the base of the sward (%) | 56 | 61 | 60 | 2.6 | NS |
| Leaf stage (number leaves) | 2.8 [a] | 3.1 [b] | 3.3 [b] | 0.05 | *** |
| Free leaf length (cm) | 24.6 [a] | 26.4 [b] | 26.6 [b] | 0.35 | *** |
| Total green leaf lamina per plant (cm) | 47.4 [a] | 58.1 [b] | 56.0 [c] | 1.33 | *** |

[1] mean data across the four target herbage masses; Different superscripts within rows denote significant differences means ($p < 0.05$). ***[2]—level of significant— *** = 0.001, NS = not significant.

### *3.2. Light at the Base of the Sward*

Light at the base of the sward was greatest ($p < 0.001$) in the low THM compared to the medium, high and very high THMs (Table 1). There was no significant effect of DD on light at the base of the sward (Table 2).

### *3.3. Leaf Stage*

There was a significant ($p < 0.001$) interaction between THM and year on leaf stage; the high herbage mass had a greater leaf stage in year 1 ($3.8 \pm 0.09$ leaves) compared to year 2 ($2.8 \pm 0.09$ leaves), and all other THMs were similar in year 1 and year 2. There was a significant interaction effect between THM and DD ($p < 0.05$) on leaf stage. The low THM in DD1 ($1.9 \pm 0.11$ leaves) had a lower leaf stage compared to DD3 ($2.6 \pm 0.11$ leaves), with DD2 ($2.2 \pm 0.11$ leaves) having an intermediate leaf stage. The medium THM had a greater leaf stage in DD3 ($3.3 \pm 0.11$ leaves) compared to DD1 and DD2 ($2.6 \pm 0.11$ leaves). The leaf stage increased significantly ($p < 0.001$) from the low THM to the very high THM (Table 1). Leaf stage was significantly ($p < 0.001$) lower in DD1 compared to DD2 and DD3 (Table 2). Year had a significant ($p < 0.001$) effect on leaf stage. Year 1 ($3.2 \pm 0.04$ leaves) had a greater leaf stage than year 2 ($2.9 \pm 0.04$ leaves).

### *3.4. Free Leaf Lamina*

There was a significant interaction ($p < 0.001$) between THM and year on the FLL, the low, medium and high THMs had a greater FLL in year 1 (21.8, 26.8 and $29.4 \pm 0.57$ cm, respectively), compared to year 2 (15.3, 23.2 and $25.3 \pm 0.57$ cm, respectively). The very high herbage mass had a similar FLL in year 1 and year 2 ($32.8 \pm 0.57$ cm). There was a significant interaction ($p < 0.001$) between DD and year on FLL; in year 1, DD2 and DD3

(28.4 and 30.4 $\pm$ 0.49, respectively) had a greater FLL compared to DD2 and DD3 in year 2 (24.5 and 22.9 $\pm$ 0.49, respectively). Free leaf lamina was significantly ($p < 0.001$) different for all THMs (Table 1). Defoliation date had a significant ($p < 0.001$) effect on FLL, and DD2 and DD3 (26.5 $\pm$ 0.35 cm) had a greater FLL than DD1 (24.7 $\pm$ 0.35 cm). Year significantly ($p < 0.001$) affected FLL, with year 1 (27.8 $\pm$ 0.29 cm) having a greater FLL compared to year 2 (24.1 $\pm$ 0.29 cm).

### 3.5. Total Green Leaf Lamina

There was a significant ($p < 0.001$) difference between all THMs for total green leaf lamina (Table 1). Defoliation date had a significant ($p < 0.001$) effect on total green leaf lamina. The total green leaf lamina increased from DD1 to DD2 to DD3 (Table 2), and all DDs were different from each other.

### 3.6. Percentage of Tillers with Internode Elongation Present

There was a significant ($p < 0.001$) effect of THM $\times$ DD on internode elongation; the percentage of tillers with internode elongation increased from DD1 to DD3 more markedly in the very high compared to the high THM, whereas the medium and low THMs maintained similar percentage values across the three defoliation dates (Table 3). The percentage of tillers with internode elongation was significantly ($p < 0.001$) affected by THM (Table 3). The medium THM had the lowest percentage, followed by the low THM and the high THM. The very high THM had the greatest percentage of tillers with internode elongation. Defoliation date significantly affected the percentage of tillers with internode elongation present (Table 3). The lowest percentage of internode elongation present was found in DD1. Year had a significant ($p < 0.001$) effect on the percentage of tillers with internode elongation; year 1 (32 $\pm$ 1.1%) had a greater percentage of tillers with internode elongation compared to year 2 (21 $\pm$ 1.1%).

**Table 3.** The effect of the interaction of target herbage mass (THM; low—500 kg DM ha$^{-1}$, medium—1500 kg DM ha$^{-1}$, high—2000 kg DM ha$^{-1}$ and very high—3000 kg DM ha$^{-1}$) and defoliation date (DD; DD1—15 October, DD2—7 November and DD3—21 November) of defoliation in autumn on the percentage of tillers (%) with visible internode elongation.

| THM | Low | Medium High | Very High | DD Average | S.E. | | Significance | |
|---|---|---|---|---|---|---|---|---|
| Percentage of Tillers with Visible Internode Elongation (%) | | | | | | THM | DD | THM $\times$ DD |
| DD1 | 18.8 | 8.9 21.2 | 30.9 | 20.0 | 2.5 | ***2 | *** | *** |
| DD2 | 16.4 | 9.2 23.8 | 51.3 | 25.2 | | | | |
| DD3 | 17.1 | 10.2 40.8 | 78.0 | 36.5 | | | | |
| THM average | 17.3 | 9.4 28.6 | 53.4 | | | | | |

***2—level of significant—*** = 0.001.

### 3.7. Length of Internode Elongation

There was a significant ($p < 0.001$) effect of THM $\times$ DD on the length of internode elongation (Table 4); THM had a significant ($p < 0.001$) effect on the length of internode of elongation (Table 4). The very high THM had a longer internode elongation compared to the low THM, with the medium and high THMs intermediate to both. Defoliation date did not significantly ($p > 0.05$) affect the length of internode elongation. Year had a significant ($p < 0.001$) effect on the length of internode elongation; year 2 (2.2 $\pm$ 0.18 cm) had a greater length of internode elongation compared to year 1 (1.0 $\pm$ 0.18 cm).

**Table 4.** The effect of the interaction of target herbage mass (THM; low—500 kg DM ha$^{-1}$, medium—1500 kg DM ha$^{-1}$, high—2000 kg DM ha$^{-1}$ and very high—3000 kg DM ha$^{-1}$) and defoliation date (DD; DD1—15 October, DD2—7 November and DD3—21 November) of defoliation in autumn on the average length of visible internode elongation (cm) and the proportion of tillers > 4 cm are presented.

| THM | Low | Medium | High | Very High | DD Average | S.E. | | Significance | |
|---|---|---|---|---|---|---|---|---|---|
| | | | | | | | THM | DD | THM × DD |
| **Length of visible internode elongation (cm) *** | | | | | | | | | |
| DD1 | 0.45 | 1.84 | 0.69 | 1.94 | 1.23 | 0.338 | ***[2] | NS | *** |
| DD2 | 0.61 | 1.63 | 0.87 | 2.19 | 1.33 | | | | |
| DD3 | 0.88 | 1.38 | 1.50 | 2.02 | 1.45 | | | | |
| THM average | 0.65 | 1.60 | 1.02 | 2.05 | | | | | |
| **Percentage of tillers with apical meristems >4 cm (%)** | | | | | | | | | |
| DD1 | 0.1 | 2.0 | 3.7 | 7.8 | 3.4 | 2.3 | *** | NS | *** |
| DD2 | 2.6 | 2.1 | 1.2 | 9.7 | 3.9 | | | | |
| DD3 | 0.4 | 0.2 | 1.4 | 13.4 | 3.8 | | | | |
| THM average | 1.0 | 1.4 | 2.1 | 10.3 | | | | | |

* Only including tillers that have internode elongation present. ***[2]—level of significant—*** = 0.001, NS = not significant.

*3.8. Percentage of Tillers with Apical Meristems > 4 cm*

There was a significant ($p < 0.001$) effect of THM × DD on the percentage of tillers with apical meristems > 4 cm (Table 4); in DD1, DD2 and DD3, the low, medium and high THMs had a lower number of tillers with apical meristems > 4 cm compared to the very high THM. THM had a significant ($p < 0.001$) effect on the percentage of tillers with apical meristems > 4 cm (Table 4). The very high THM had a greater number of tillers with apical meristems > 4 cm compared all other THMs. Defoliation date and year did not significantly affect the percentage of tillers with apical meristems > 4 cm.

## 4. Discussion

The current study shows that autumn grazing management strategies, i.e., accumulating varying THMs and different DDs, have significant impacts on sward morphology (leaf stage, FLL and presence and length of internode elongation). Practically, this indicates that some flexibility might be required if growth rates are greater than usually predicted (~15 kg DM/ha/day) [27]. Previous studies have reported that, to ensure optimal grazing utilisation, swards should be grazed at the three-leaf stage [18] and at a herbage mass of 1500 kg DM ha$^{-1}$ [19]. In the current study, the low THM did not meet the three-leaf stage (DD1—1.9 leaves) and herbage mass (900 kg DM ha$^{-1}$) requirements to ensure optimum utilisation for grazing. However, low THM had the greatest percentage of light transmitted to the base of the sward in autumn, and as a result, it did not trigger an increased amount of internode elongation compared to the high and very high THMs. Previous studies have reported that light transmitted to the base of the sward impacts tiller appearance [26] and internode elongation [32]. Decreasing light to the base of the sward can result in tiller bud suppression [33] which can result in the loss of daughter tillers [34] and can inhibit the production of new tillers [35,36]. The light transmitted to the base of the sward decreased (−9%) from the low to the medium THM, which did not increase the % of tillers with visible internode elongation, but did increase the length of visible internode elongation between the low and medium THMs (+ 0.95 cm) [32]. This indicates that although the light at the base of the sward did decrease between the low and medium THMs, the number of tillers with visible internode elongation did not change significantly. However, as shading increased, the length of the internode elongation in tillers also increased. Internode elongation is an unfavourable morphological trait in PRG swards as the growing point is no longer at ground level [37]. If internode elongation is above the grazing/defoliation height

(3.5 cm in the current study), it could result in increased tiller mortality after defoliation. This can then lead to a reduction in growing points [27] for herbage production following defoliation. In the current study, when swards reached a herbage mass of between ~1700 kg DM ha$^{-1}$ and ~2900 kg DM ha$^{-1}$, the light transmitted to the base of the sward remained constant at 57%, similar to Thomas and Norris [35]. The medium and high THMs had increased effects on internode elongation; however, the greatest impact was seen in the very high THM, with 53% of tillers having internode elongation, compared to 29% in the high THM, and a greater percentage of internode elongation above 4 cm. The delay between defoliations (DD1–DD2–DD3) did not impact the amount of light transmitted to the base of the sward as it had reached its saturation point for light transmission. The light transmitted to the base of the sward in the current study did not decrease when herbage mass increased (1735 kg DM ha$^{-1}$ to 2917 kg DM ha$^{-1}$). However, increasing the length of time between defoliations in autumn did increase the percentage of tillers with visible internode elongation.

Post defoliation, all plots were found to return to approximately 79 ± 2.9% light at the base of the sward in autumn, and there was no effect on tiller density [30]. This increased light transmission to the base of the sward [38] would have stimulated the production of tillers [26,39], giving enough growing points for production [27] in spring. The very high THM experienced the greatest morphological changes over the experimental period due to the length and level of shading [26]. To reduce prolonged shading and the impacts on internode elongation, the very high THM should be defoliated as early as possible in autumn in an on-farm scenario. Defoliation should ideally happen in swards when they are at a medium THM (1500–1700 kg DM ha$^{-1}$), as this is the correct leaf stage and will allow light transmission to the base of the sward, promoting new tiller production [26,39].

The leaf stage increased as THM increased from a low THM (2.2 leaves) to a very high THM (3.9 leaves; Table 1), similar to Fulkerson and Donaghy [18], with any increase in leaf stage past the three-leaf stage leading to the senescence of the fourth leaf [20,21]. This can contribute to a decrease in herbage mass in swards that go past the three-leaf stage [31] as a result of an increase in senescent material, which reduces sward quality [40], due to the lower level of organic matter in dead material [41]. Although there was a reduction in herbage and an increase in senescent material from early November in the very high THM, the FLL and green leaf lamina available for grazing were greatest in the very high THM. A greater FLL was identified as being beneficial in perennial ryegrass breeding as it can result in increased grazing efficiency due to the greater amount of green leaf available for defoliation [42]. However, in the current study, the greatest FLL was found in the very high THM, accompanied by a high level of senescent material. This would reduce grazing efficiency, as there is an increased level of senescent material of lower quality [40] being grazed. It was found that the FLL never surpassed approximately 33 cm, and this appeared to be the maximum achievable FLL in autumn. After the maximum is reached, the amount of senescent material will continue to increase, and this will lead to reduced sward quality [40].

## 5. Conclusions

The current study reported that the light transmitted to the base of the sward and sward morphology are impacted by autumn grazing management for the extension of the grazing season. Management practices can be altered to achieve grazing targets and improve sward morphology. In swards with a low THM, enough light is transmitted to the base of the sward, but they do not reach the optimal three-leaf stage or herbage mass for grazing and should be defoliated from the middle of November. Defoliation can decrease the effect of reduced light transmitted to the base of the sward on internode elongation, as all swards, regardless of herbage mass, return to the same light interception to the base of the sward post defoliation. Swards from the medium THM to the very high THM experience reduced light transmission to the base of the sward prior to defoliation. Ideally, swards with a medium THM at DD1 are ideal for optimum utilisation; however, for the

extension of the grazing season, high and very high THMs are necessary. To counter the negative effects of accumulating high and very high THMs, defoliation of the high and, in particular, the very high THM between mid-October and early November was found to benefit the sward, as internode elongation will be reduced. Earlier defoliation will also reduce the number of tillers with internode elongation > 4 cm in the very high THM.

**Author Contributions:** Conceptualization, M.E.; methodology, M.E., C.L. and D.D.; formal analysis, M.E. and C.L.; investigation, C.L.; resources, M.E.; data curation, C.L.; writing—original draft preparation, C.L.; writing—review and editing, M.E., D.D. and A.W.; supervision, M.E. and A.W.; project administration, M.E.; funding acquisition, M.E. All authors have read and agreed to the published version of the manuscript.

**Funding:** This experiment was funded by the Irish Dairy Levy Funding administered by Dairy Research Ireland and the Teagasc Walsh Scholarship programme.

**Data Availability Statement:** Data are contained within the article.

**Acknowledgments:** The authors wish to thank D. Hennessy and M. O. Donovan for their valuable input on the experiment; M. Liddane, A. McGrath, T. Casey and P. O. Connor for their technical assistance; and all the staff and students of Moorepark research farm for their assistance with measurements during the experiment.

**Conflicts of Interest:** The authors declare no conflicts of interest.

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
