# Peer review of "The Impact of Accumulating Herbage Masses in Autumn on Perennial Ryegrass Sward Characteristics"

_agronomy, doi:10.3390/agronomy14010148_

Round 1

Reviewer 1 Report

Comments and Suggestions for Authors

There are some points that the authors need to be careful and correct, the work has relevance and merit to be considered for publication

line 24: replace the words: autumn and forage mass as it is in the title.

Introduction: The hypothesis of your precise work is clear. There is an excess of information in the literature, but the authors escape the central hypothesis of the work.

Line 115: Was the defoliation carried out by an animal? If done manually, this must be declared.

Results

Did the authors perform chemical composition analysis? I imagine that working with cutting times and times of the year, there are fluctuations in crude protein and fiber.

At the end of the discussion, the authors need to make a conclusion, preparing the author to prepare the readers for the conclusion.

Conclusion:

Synthesize and improve the conclusion, making a final recommendation.

Reviewer 2 Report

Comments and Suggestions for Authors

Dear Editor and Authors, after reading the manuscript agronomy-2726998, Title "The impact of accumulating herbage masses in autumn on perennial ryegrass sward characteristics", follows the considerations of the present reviewer:

The article investigated the impact of different target herbage masses in autumn and imposed different defoliation dates between mid-October and late November on forage canopy structure parameters. The topic is relevant to drive efficient pasture management. The manuscript has scientific merit.

ABSTRACT (lines 11-23): Generally written satisfactorily. In lines 28-29 the authors describe the importance of the forage plant studied specifically in  Ireland and New Zealand, but is it only in these countries? If so, what is the reason/relationship for a plant to be so important only in two geographically distant and different countries? Authors must include the significance of the test that generated the results presented (p-value).

INTRODUCTION (lines 27-81): Remove or change one of the sentences “previous studies”, repeated in lines 46 and 51). Describe the scientific hypothesis that supported this work.

MATERIAL AND METHODS (lines 82-182): The number of repetitions, experimental design, methods for quantifying variables, and data analysis were satisfactorily described. The experimental methods are suitable for the proposed experiment. However, the present reviewer's concern is related to the experimentation time. Usually, two (02) years (two climatic seasons) are satisfactory for obtaining/validating results in this type of study. However, as stated in the title and objective of the study, it is a perennial crop. In this case, the present evaluator is unsure about the inference of the two-year results on the inference regarding the real perpetuity and impacts on the forage plant, under the evaluated management conditions.

RESULTS (lines 183-274): In Tables 2, 3, and 4 (lines 195, 201, 246, and 270), replace * with the actual p-values. Table 4 should not be divided into “A” and “B”, but rather into Table 4 and Table 5.

DISCUSSION (lines 275-339): The results could be better explored and discussed, including explanations and practical applications of experimental findings on pasture management. Based on the results observed, what would be the authors' implications and recommendations for efficient pasture management? What could change in relation to what is usually already done?

CONCLUSIONS (lines 341-354): The present evaluator suggests changing the conclusion. The text in this section has discussion segments and implications. I suggest making the conclusion more objective, focusing on the main results observed.

Reviewer 3 Report

Comments and Suggestions for Authors

The manuscript agronomy-2726998 presents a good experimental work on the autumn management of perennial ryegrass pastures, examining the effects of different accumulated herbage masses and defoliation dates on sward agronomic characteristics. It provides relevant results to adopt the best management strategies for a more efficient use of autumn pastures by grazing animals. However, there are some flaws that need to be properly addressed before the manuscript can be published as a meritorious scientific article.

First of all, although the experimental design seems appropriate, it is not fully described regarding the (nesting?) structure of the plots-split plots. It follows that the statistical model applied to data must be consistent with that design.

The results in general are well described but the meaning of the significant interactions is not clear. Finally, there are many incomplete references that make it extremely difficult to evaluate their power to support the rationale of the experiment and the results obtained.

Below there are specific points that should be taken into account to improve the scientific quality of the manuscript.

1) L17-21: In the whole abstract only results on light interception and internode elongation are provided. Although the abstract is near the word limit (200 words), I suggest adding some result (the most important) on other measured variables such as leaf stage or lamina length.

2) L17-18: “Light transmitted to the base of the sward … decreased from the medium THM to the very high THM.” This is not entirely true as there were no differences between medium, high and very high THM according to Table 2. Please correct the sentence.

3) L96: As meteorological data is available, could the authors comment on it? Were the recorded parameters in the experimental years normal compared to the averages? Were the two years similar or different between them? Could such differences cause the between-year differences observed in sward characteristics? In any case, the latter would be for discussion.

4) L102: How was the structure of the split plot design? Which factor was assigned to whole plots or blocks and which to split plots? Was a fixed factor nested into the other? This should be clearly explained as the applied statistical model depends on it (see comment below).

5) L161: “The total green leaf lamina length was also recorded.” Was this achieved by adding up the length of all present (non-senescent) leaves?

6) L173-175: The statistical model looks strange. There is an interaction term of year with DD but not of year interacting with THM. In fact, significant interactions between THM and year are commented later on (L211, 223). The interaction between the two fixed effects appears twice, but in different ways (with swapped subscripts). If j denotes the THM levels (j = 1, 2, 3, 4) and k the DD levels (k = 1, 2, 3), then applying j to DD and k to THM in the last interaction term seems wrong. As the experimental design is not fully described, it is difficult to assess the adequacy of the applied model.

7) L182: Was a post-hoc test used to establish differences among means? Which one (i.e. Tukey, Bonferroni…)?

8) L184: The sub-section 3.1 has no sense if there is no sub-section 3.2. Are the results from spring period missing? Since measurements were performed also in (winter)-spring, are those results not considered for this article? Rearrange and renumber the Results sub-sections accordingly.

9) L186-188: The reason for the significant interaction between DD and year is not clear. Was herbage mass at DD2 and DD3 greater in year 1 than in year 2, but similar or lower at DD3?

10) L195-197: The title of Table 2 seems wrong as only the effects of THM are shown.

11) L208-209: The values of light percentage seem wrong. The overall mean between years 1 and 2 averages should be near 59% according to the values shown in Tables 2 and 3.

12) L214-217: Again the reason for the significant interaction between THM and DD is not entirely clear. It seems that there is a general pattern of increasing leaf stage from DD1 to DD3, but this might not occur in some THMs, leading to a significant interaction between the two factors. Could this be the case? As only the main effects of the fixed factors (THM in Table 2, DD in Table 3) are shown for this dependent variable (I assume that there was no significant THM x DD interaction in the case of the other four variables), the nature of the interaction should be clearly explained in the text. Alternatively, a figure showing the 4 x 3 least square means ± standard errors could be added.

13) L233: There is no comment on the results about total green leaf lamina per plant. As main results are shown in Tables 2 and 3, those should be briefly described in text. In general, their correlation with FLL results is high, but no so much in the case of between-DD differences.

14) L236-237: The nature of THM x DD interaction could be better described as follows (it is just a suggestion): “The percentage of tillers with internode elongation increased from DD1 to DD3 more markedly in the very high compared to the high THM, whereas the medium and low THM maintained similar percentage values across the three defoliation dates.”

15) L253-254: This could also be applied to DD2. In the interaction the different pattern between THMs across the DDs should be explained. Suggestion: “Delaying the DD caused an enlargement in internode elongation in the low and high THM but a shortening in the medium THM, while in the very high THM the greatest length was observed at DD2.”

16) L263-265: This is not even true as in DD2 the lowest percentage occurred in the high THM. Please explain correctly the nature of the THM x DD interaction.

17) L341-354: Could the authors conclude with a particular combination of THM and DD being the best one in practice to achieve the highest utilization efficiency in their conditions?

Minor points and misprints:

L52: 1500 kg (insert blank)

L86: The reference [38] should not follow [29] (L75).

L91: Could the authors specify the PRG variety used in the study? The agronomic characteristics could be different depending on it.

L110: Delete comma after “characteristics”

L132: Add city and country for LI-COR Inc.

L144: Delete “in” before “at”

L161, 302: Replace the references with their corresponding numbers in square brackets. Add the references in the list.

L199, 204: Add the meaning of superscripts in a footnote.

L234, 251, 261: Maybe these three related sub-sections could be joined in one.

L246, 270: The numbering of tables could be 4 and 5 instead of 4A and 4B. Or perhaps they could be joined as a unique table if not too large.

L294-295: Please correct the incomplete sentence.

References: Complete all the references in the list following the instructions to authors.

Table 1: Please add the units for solar radiation. Be consistent in the number of decimal places in each dependent variable.

Table 4B: Delete “Average” in the first variable (“Length of visible…”) as all the values shown are in fact averages. I suggest changing the second variable to “Percentage of tillers with apical meristems > 4 cm (%)”. Be consistent in the number of decimal places (DD averages for the second variable).

Comments on the Quality of English Language

In general the quality of English is good.

Round 2

Reviewer 3 Report

Comments and Suggestions for Authors

The manuscript agronomy-2726998 has been improved following the suggestions of the reviewers, but some points have not been fully addressed or remain uncorrected.

Experimental design: It is better described now, but I think the statistical model (L176-179) is still wrong as it is written; the error term should be an independent one, i.e., not multiplying the preceding interaction, and the 3-way interaction is missing. In my opinion, if the full model with all possible interactions among factors was tested (regardless of whether it is later decided to remove non-significant interactions from the model), it could be written as follows (note that using abbreviations for the fixed effects makes the math model more readable): Y_jkl = μ + THM_j + DD_k + year_l + THM_j × DD_k  + year_l × THM_j  + year_l × DD_k  + year_l × THM_j  × DD_k  + e_jkl

However, I am not sure if the split plot design is well reflected with this model. In fact, there could be two error terms, one for the whole plots and one for the split plots. I suggest that authors consult with a proficient statistician to display the correct model.

L187. Please add an explanation on the test used to differentiate the group means (Tukey).

L203, 208. Change table footnotes to: “Different superscripts within rows denote significantly different means (P < 0.05).”

L251-253. The suggested text “delaying the DD caused an enlargement in internode elongation in the low and high THM but a shortening in the medium THM, while in the very high THM the greatest length was observed at DD2.” has been inserted in the wrong place as it refers to the next subsection 3. 7 Length of internode elongation. It should be moved to L264-265 replacing the text “within DD3 … intermediate to both”

L50. 1500 kg (insert a blank between number and units)

Table 1. Add a decimal place to 6 for Average daily air temp in Dec Year 2.

Table 5. Please check the DD average values for % tillers. Are the three values (4.0) so coincident? Averaging the four values for each THM, the respective values for DD1, DD2 and DD3 would be 3.4, 3.9 and 3.8.

Follow the instructions to authors for references, e.g. abbreviated journal names (L403, 407, 415, 438, 449, 450, 453, 461, 462, 467, 470), publication year (L402 and following), etc.

L403. Add missing volume number (50).

L442. Add missing volume and page numbers.

L450. Add missing volume number (80).

L456. Delete quotes.

L462. Add missing volume number (62).

Comments on the Quality of English Language

In general the quality of English is good, although some editing might be required.

Author Response

Second revision: Reviewer 3

The manuscript agronomy-2726998 has been improved following the suggestions of the reviewers, but some points have not been fully addressed or remain uncorrected.

Experimental design: It is better described now, but I think the statistical model (L176-179) is still wrong as it is written; the error term should be an independent one, i.e., not multiplying the preceding interaction, and the 3-way interaction is missing. In my opinion, if the full model with all possible interactions among factors was tested (regardless of whether it is later decided to remove non-significant interactions from the model), it could be written as follows (note that using abbreviations for the fixed effects makes the math model more readable): Y_jkl = μ + THM_j + DD_k + year_l + THM_j × DD_k  + year_l × THM_j  + year_l × DD_k  + year_l × THM_j  × DD_k  + e_jkl

However, I am not sure if the split plot design is well reflected with this model. In fact, there could be two error terms, one for the whole plots and one for the split plots. I suggest that authors consult with a proficient statistician to display the correct model.

Author response: The model has been updated as per the first comment. The authors have consulted a statistician to discuss the above comment. After meeting and describing the design, the statistician provided comments and feedback on the design. They advised that the split plot should not have been included in the design or stats section as it was not a true split plot design. This has now been amended in the experimental design section and the stats model now appropriately matches the experimental design

L187. Please add an explanation on the test used to differentiate the group means (Tukey).

Author response: Explanation of test added in L169-171

L203, 208. Change table footnotes to: “Different superscripts within rows denote significantly different means (P < 0.05).”

Author response: Corrected as per reviewers comments in L203, 208.

L251-253. The suggested text “delaying the DD caused an enlargement in internode elongation in the low and high THM but a shortening in the medium THM, while in the very high THM the greatest length was observed at DD2.” has been inserted in the wrong place as it refers to the next subsection 3. 7 Length of internode elongation. It should be moved to L264-265 replacing the text “within DD3 … intermediate to both”

Author response: Updated within the manuscript lineL260-262

L50. 1500 kg (insert a blank between number and units)

Author response: Blank inserted between number and units as requested.

Table 1. Add a decimal place to 6 for Average daily air temp in Dec Year 2.

Author response: Decimal point added, now 6.0

Table 5. Please check the DD average values for % tillers. Are the three values (4.0) so coincident? Averaging the four values for each THM, the respective values for DD1, DD2 and DD3 would be 3.4, 3.9 and 3.8.

Author response: Apologies, I believe the table just updated to four and I added in the decimal points manually. Updated within the document as recommended.

Follow the instructions to authors for references, e.g. abbreviated journal names (L403, 407, 415, 438, 449, 450, 453, 461, 462, 467, 470), publication year (L402 and following), etc.

L403. Add missing volume number (50).

L442. Add missing volume and page numbers.

L450. Add missing volume number (80).

L456. Delete quotes.

L462. Add missing volume number (62).

Author response: Referencing updated with volume and page numbers where possible. No volume available for reference number 21.
